# Planktonic foraminifera organic carbon isotopes as archives of upper ocean carbon cycling

Babette A. A. Hoogakker ®[1] ✉, Caroline Anderson[2], Tommaso Paoloni ®[1], Andrew Stott[3], Helen Grant[3], Patrick Keenan[3], Claire Mahaffey[4], Sabena Blackbird ®[4], Erin L. McClymont ®[5], Ros Rickaby ®[6], Alex Poulton ®[1] & Victoria L. Peck[7]

The carbon cycle is a key regulator of Earth's climate. On geological time-scales, our understanding of particulate organic matter (POM), an important upper ocean carbon pool that fuels ecosystems and an integrated part of the carbon cycle, is limited. Here we investigate the relationship of planktonic foraminifera-bound organic carbon isotopes ($\delta^{13}C_{org\text{-}pforam}$) with $\delta^{13}C_{org}$ of POM ($\delta^{13}C_{org\text{-}POM}$). We compare $\delta^{13}C_{org\text{-}pforam}$ of several planktonic for-aminifera species from plankton nets and recent sediment cores with $\delta^{13}C_{org\text{-}POM}$ on a N-S Atlantic Ocean transect. Our results indicate that $\delta^{13}C_{org\text{-}pforam}$ of planktonic foraminifera are remarkably similar to $\delta^{13}C_{org\text{-}POM}$. Application of our method on a glacial sample furthermore provided a $\delta^{13}C_{org\text{-}pforam}$ value similar to glacial $\delta^{13}C_{org\text{-}POM}$ predictions. We thus show that $\delta^{13}C_{org\text{-}pforam}$ is a promising proxy to reconstruct environmental conditions in the upper ocean, providing a route to isolate past variations in $\delta^{13}C_{org\text{-}POM}$ and better under-standing of the evolution of the carbon cycle over geological time-scales.

In the open ocean, suspended particulate organic matter (POM) fuels marine ecosystems. POM is a composite pool of material, including phytoplankton, zooplankton, bacteria, and detritus, with the relative proportion of each varying as a function of ecosystem structure. POM is an important component of the carbon cycle, with the export and burial of POM in marine sediments representing a major carbon sink on geological time-scales. Long-term (>100 Ma) changes in productivity and burial of POM can modify atmospheric carbon dioxide and oxygen concentrations[1].

Sizeable changes in the POM pool or burial of POM in sediments over time can alter the carbon isotopic composition of dissolved inorganic carbon ($\delta^{13}C_{DIC}$) in seawater. Seawater $\delta^{13}C_{DIC}$ is a key diagnostic for ocean circulation and cycling of carbon between the oceans

and land, with past variations in $\delta^{13}C_{DIC}$ inferred from the $\delta^{13}C$ of carbonate of specific benthic foraminifera[2]. Changes in the burial ratio of organic and inorganic carbon can drive changes in $\delta^{13}C_{DIC}$ on time-scales >0.3–1 Ma[3,4]. On shorter, glacial-interglacial timescales, changes in the amount of respired organic carbon (remineralized POM) in the global ocean[5–8], can influence $\delta^{13}C_{DIC}$[9,10]. Changes in the carbon isotopic composition of POM ($\delta^{13}C_{org\text{-}POM}$), have so far not been considered as a driver of change in $\delta^{13}C_{DIC}$, but on timescales >20 kyrs, $\delta^{13}C_{DIC}$ may be sensitive to changes in $\delta^{13}C_{org\text{-}POM}$.

In the modern ocean, $\delta^{13}C_{org}$ of net collected plankton and suspended POM varies from −20 ± 2‰ at low to mid latitudes, to −30 ± 4‰ in the Southern Ocean, controlled by fractionation during photosynthesis[11,12]. The degree of isotopic fractionation relates to the

[1]The Lyell Centre, Heriot-Watt University, Edinburgh, UK. [2]School of Geography and the Environment, University of Oxford, South Parks Road, Oxford OX1 3QY, UK. [3]UK Centre for Ecology and Hydrology, Lancaster LA1 4AP, UK. [4]School of Environmental Sciences, University of Liverpool, 4 Brownlow Street, Liverpool L69 3GP, UK. [5]Department of Geography, Durham University, Lower Mountjoy, Durham DL1 3LE, UK. [6]Department of Earth Sciences, University of Oxford, South Parks Road, Oxford OX1 3AN, UK. [7]British Antarctic Survey, High Cross, Madingley Road, Cambridge CB3 0ET, UK. ✉e-mail: b.hoogakker@hw.ac.uk

availability of nutrients and light, temperature and pH conditions, carbon concentration mechanisms, and growth rates[13–15], as well as the relative abundance of autotrophs (phytoplankton), heterotrophs (bacteria, zooplankton) and detrital material.

Plankton community structure and POM composition dictate the strength of the relationship between $\delta^{13}C_{org\text{-}POM}$ and the concentration of $CO_{2aq}$; a strong relationship is observed in productive eutrophic communities where autotrophs dominate, whereas a poor relationship is observed in oligotrophic recycling communities, where heterotrophic and detrital material dominate the POM[13,16–18]. In the Atlantic Ocean, ¹³C-depleted POM is found in cold high-latitude waters with high $CO_2$ concentrations (and high production); with the most ¹³C depleted POM values found south of 50° S (Fig. 1)[12,15,19]. Near the Antarctic coast, plankton communities change and sea-ice can cause highly variable suspended $\delta^{13}C_{org\text{-}POM}$[18,20,21]. ¹³C-enriched POM at -10° N (Fig. 1) in the North Atlantic has been attributed to reduced isotopic fractionation at high growth rates within this warm, high productivity area, fuelled in part by equatorial upwelling[22].

While we have some knowledge about modern $\delta^{13}C_{org\text{-}POM}$, little is known about past $\delta^{13}C_{org\text{-}POM}$. For example, during ice ages/glacial stages, lower atmospheric $CO_2$ concentrations and colder sea surface temperatures are thought to have caused a reduction in air–sea fractionation, which in turn would have caused enrichment in $\delta^{13}C_{org\text{-}POM}$[23]. Bulk sediment $\delta^{13}C_{org}$ measurements from the Atlantic over the past glacial–interglacial cycle, however, do not show uniform enrichment[23–26]. This suggests that locally either other factors play a role, or that processes other than $pCO_2$/temperature can overprint the signal in bulk sediments[27]. Here we assess whether planktonic foraminifera-bound $\delta^{13}C_{org}$ ($\delta^{13}C_{org\text{-}pforam}$) provides a better representation of suspended $\delta^{13}C_{org\text{-}POM}$. We focus on a N-S Atlantic Ocean transect, using living and recent (Holocene) planktonic foraminifera. We also compare our Atlantic results with a recent study by Swart et al.[28] from across the tropical equatorial Pacific.

Planktonic foraminifera are single-celled eukaryotic microorganisms with calcite tests that can be found in almost all open-ocean environments, from polar to equatorial waters, in both eutrophic and oligotrophic waters. The life cycle of different species varies from two to four weeks (shallow mixed-layer dwelling species) to a year (deeper dwelling species). Planktonic foraminifera are omnivorous; their diet varies between species and habitat; but includes bacteria, phytoplankton, zooplankton, and in some cases other foraminifera. Spinose species prefer zooplankton prey, including large metazoans, whereas non-spinose species are mainly herbivorous, though they may also ingest detrital particulate material[29]. Some planktonic foraminifera bear symbionts: the metabolic carbon of these may cause enrichment in $\delta^{13}C_{org\text{-}pforam}$[30].

Secretion of calcium carbonate crystal lattices to form the test chambers takes place on an extremely thin (0.05–0.06 μm) primary organic membrane that is produced by strands of the cell's cytoplasm[29]. In living planktonic foraminifera, the cytoplasm (cellular organic material) weighs about 2.8 times that of the test calcite[31]. The living planktonic foraminifera analysed here still contained their cytoplasm. Cytoplasm is lost from the calcite test following gametogenesis; the shells that subsequently sink to the ocean floor and are preserved in sediments are void of cytoplasm.

Within deep-sea sediments, degradation of the organic material takes place by aerobic and anaerobic processes. The organic membrane, contained within the walls of the foraminifera test, may be physically protected, though diagenesis during burial has been associated with a 65–75% reduction in preserved proteins[32,33]. The remaining test-bound organic membrane, typically only 0.1% of calcite test weight[34], is the organic material associated with foraminifera buried in sediments.

Foraminifera calcite is arguably one of the most useful and commonly analysed materials for palaeoceanographers. Foraminifera calcite chemistry enables oxygen isotope stratigraphy, radiocarbon dating, watermass tracing and insights into nutrient cycling and oceanic variability in carbonate chemistry (pH), temperature and salinity[35]. The chemistry of the organic material contained within fossil tests, assumed to be mainly amino acids[33], is less well studied. In a culturing study, Ní Fhlaithearta et al.[36] found that metabolic carbon is the main carbon source fixed within benthic foraminifera organic linings. This organic matter may be isolated using standard decalcification techniques[37], an approach used in palaeoceanographic studies to

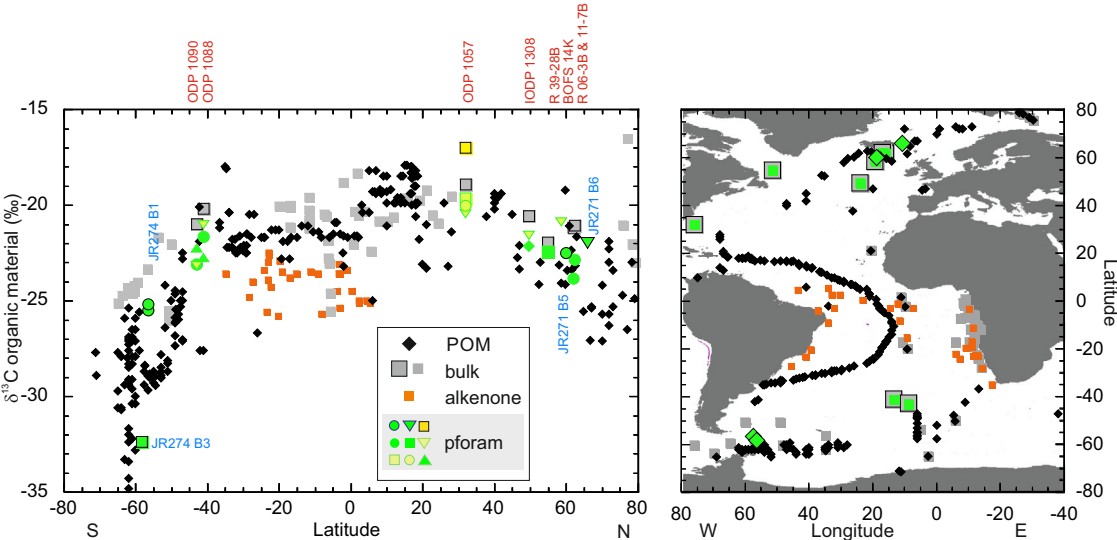

**Fig. 1 | Main results.** Right: sample locations of surface water $\delta^{13}C_{org\text{-}POM}$ (black diamonds[45]), bulk sediments (grey squares[20,28,48–51]), alkenone $\delta^{13}C$ (orange squares[52]) and planktonic foraminifera $\delta^{13}C_{org}$ (green diamonds plankton net; green squares core tops). Core top alkenone $\delta^{13}C$ are restricted to the South Atlantic; data from areas characterized by lateral transport (e.g. ref. 56) are not included. Left: $\delta^{13}C_{org}$ of surface water particulate organic matter (POM, black diamonds), sedimentary organic matter (grey squares; grey squares with black border, this study), alkenones (orange squares[52]) and planktonic foraminifera (pforam): green (black border): plankton nets: triangle−T. quinqueloba, circles−G. bulloides, square−N. pachyderma; core tops: yellow square−T. sacculifer; green circles−G. bulloides, green square−N. pachyderma, green lined yellow triangle (pointing down)−G. inflata, green lined yellow square−N. dutertrei, green lined yellow circle−P. obliquiloculata; green triangle (pointing up)−G. truncatulinoides. Sample names of plankton tow and core tops are written in blue and red (RAPiD abbreviated to R).

**Table 1 | Site location details**

|  | Latitude | Longitude | Water depth (m) |
|---|---|---|---|
| **Plankton net** | | | |
| JR271 B5 | 60.00 | −18.67 | 200 m |
| JR271 B6 | 65.97 | −10.72 | 200 m |
| JR274 B1 | −56.47 | −57.42 | 200 m |
| JR274 B1 | −56.47 | −57.42 | 200 m |
| JR274 B3 | −58.37 | −56.25 | 200 m |
| **Core top** | | | |
| RAPiD 06-3B | 62.06 | 16.06 | 2228 |
| RAPiD 11-7B | 62.30 | 17.15 | 2126 |
| BOFS 14 K | 58.62 | 19.43 | 1756 |
| RAPiD 39-28 B | 54.91 | 51.48 | 2863 |
| 303_306 U 1308 | 49.53 | 24.14 | 3871 |
| 172 1057 | 32.03 | 76.02 | 2584 |
| 177 1088 | −41.14 | 13.56 | 2082 |
| 177 1090 | −42.91 | 8.90 | 3702 |

**Table 2 | Main results; bulk sediment $\delta^{13}C_{org}$, planktonic foraminifera species, $\delta^{13}C_{org}$ (core top and plankton net)**

|  | Latitude | Bulk sediment $\delta^{13}C_{org}$ | Planktonic foraminifera species | Planktonic foraminifera $\delta^{13}C_{org}$ |
|---|---|---|---|---|
| **Plankton net** | | | | |
| JR271 B5 | 60.00 | N/A | G. bulloides | −22.51 |
| JR271 B6 | 65.97 | N/A | T. quinqueloba | −21.97 |
| JR 274 B1 | −56.47 | N/A | G. bulloides | −25.49 |
| JR 274 B1 | −56.47 | N/A | G. bulloides | −25.17 |
| JR 274 B3 | −58.37 | N/A | N. pachyderma | −32.34 |
| **Core top** | | | | |
| RAPiD 11-7B | 62.30 | −21.05 | G. bulloides | −22.85 |
| RAPiD 06-3B | 62.06 | −21.15 | G. bulloides | −23.84 |
| BOFS 14 K | 58.62 | N/A | G. inflata | −20.86 |
| RAPID 39- 28B | 54.91 | −21.92 | N. pachyderma | −22.43 |
| 303_306 U 1308 | 49.53 | −20.56 | G. inflata | −21.56 |
|  |  |  | G. hirsuta | −22.16 |
| 172 1057 | 32.03 | −18.90 | G. inflata | −20.46 |
|  |  |  | N. dutertrei | −19.65 |
|  |  |  | P. obliquiloculata | −20.02 |
|  |  |  | G. sacculifer | −17.02 |
| 177 1088 | −41.14 | −20.16 | G. bulloides | −21.65 |
|  |  |  | G. inflata | −21.03 |
|  |  |  | G. truncatulinoides | −22.73 |
| 177 1090 | −42.91 | −20.98 | G. bulloides | −23.10 |
|  |  |  | G. inflata | −23.16 |
|  |  |  | G. truncatulinoides | −22.26 |
| **LGM sample** | | | | |
| 177 1088 33–34 cm | −41.14 | N/A | G. bulloides | −18.95 |

The ODP 1088 LGM sample is from 33–34 cm down core.

investigate the marine nitrogen cycle[38] and reconstruct paleo $pCO_2$[28,34,39].

Isolated planktonic foraminifera $\delta^{13}C_{org}$ may provide a better technique to study changes in concurrent $\delta^{13}C_{org\text{-}POM}$ compared with bulk sediment or compound-specific $\delta^{13}C_{org}$; planktonic foraminifera, the sole source of the organic material analysed for $\delta^{13}C_{org\text{-}pforam}$, can be easily separated from bulk sediments. Early work by Sackett et al.[13] and Degens et al.[12] suggested no appreciable fractionation is expected between the $\delta^{13}C_{org}$ of heterotrophic plankton and their autotrophic prey. This was supported by McCutchan et al.[40] who showed only minor enrichments in $\delta^{13}C$, up to 1‰ per trophic level. If only relatively small changes in $\delta^{13}C_{org}$ occur through trophic interactions, it follows that the $\delta^{13}C_{org}$ of a consumer can be used to determine the $\delta^{13}C_{org}$ of the original dietary source[41]. Here we test the assumption of minimal fractionation in $\delta^{13}C_{org}$ between planktonic foraminifera and their diet (i.e. suspended POM), by comparing our new planktonic foraminifera–bound $\delta^{13}C_{org\text{-}pforam}$ data with that of the $\delta^{13}C_{org\text{-}POM}$ pool (data collated by Schmittner et al.[42]). In addition, we analysed a glacial planktonic foraminifera sample to assess the application of this technique on planktonic foraminifera samples that have undergone burial.

## Results

### $\delta^{13}C_{org}$ of POM, alkenones, bulk sediments and planktonic foraminifera

Sample details and our results are shown in Fig. 1 and Tables 1 and 2, together with published $\delta^{13}C_{org\text{-}POM}$ data (source: ref. 42), bulk sediment data (source: refs. 13, 22, 43–46), and alkenone $\delta^{13}C_{org}$ data (source: ref. 47). Alkenones are long-chain ketones synthesized by some haptophyte algae[48,49] and offer an insight into the $\delta^{13}C_{org}$ signal of selected phytoplankton (i.e. foraminifera prey). Alkenones are fractionated 4.5–10‰ lighter relative to cellular biomass[50]. It should be noted that while all foraminifera samples used in this study are generally recent/Holocene, there are variations in sampling date and the ages the samples represent. Specifically, plankton-tow foraminifera samples were collected in 2012 and 2013, whereas $\delta^{13}C_{org\text{-}POM}$ samples were taken between 1960 and 2010[42]. Bulk sediment and core-top foraminifera samples typically represent averaged signals representing intervals spanning 500–1500 years.

In Fig. 2A, we compare averaged Atlantic latitudinal transects of $\delta^{13}C_{org\text{-}POM}$, bulk core top $\delta^{13}C_{org}$ and $\delta^{13}C$ data of sedimentary alkenones, while in Fig. 2B we compare our $\delta^{13}C_{org\text{-}pforam}$ with $\delta^{13}C_{org\text{-}POM}$. The comparison with alkenone data is limited to the equatorial and tropical to subtropical South Atlantic, while $\delta^{13}C_{org\text{-}pforam}$ data are

limited to subtropical to temperate and subpolar regions. A large portion of the alkenone and bulk $\delta^{13}C_{org}$ data are from the Southwest African margin, where there is only limited $\delta^{13}C_{org\text{-}POM}$ data available.

Limited $\delta^{13}C$ data of sedimentary alkenones (equator to 40° South) show that they are typically depleted (−25.8‰ to −22.5‰) and more variable compared to $\delta^{13}C_{org\text{-}POM}$ (−22.8‰ to −20.5‰, Figs. 1 and 2A). Between 10° N and 30° S, the average difference between $\delta^{13}C$ of sedimentary alkenones and $\delta^{13}C_{org\text{-}POM}$ is −2.7 ± 1.3‰ ($n = 5$), and the difference between bulk sediment $\delta^{13}C_{org}$ and $\delta^{13}C$ of alkenones is 3.7 ± 1.4‰ ($n = 4$). There could be a potential bias in the South Atlantic when comparing $\delta^{13}C_{org\text{-}POM}$ with bulk sediments and alkenones, especially when the bulk sediments have a high sample density along the west African margin where there is no POM data. When we focus our comparison on $\delta^{13}C_{org\text{-}POM}$ versus sediments (note there is only one datapoint per bin here) and alkenones from the open South Atlantic between 10° N and 30° S, then the difference between $\delta^{13}C_{org\text{-}POM}$ and bulk sediments increases by on average 1‰. This is while both open ocean $\delta^{13}C_{org\text{-}POM}$ and sediment $\delta^{13}C_{org}$ are on average enriched compared with continental margins here. Comparison of $\delta^{13}C_{org\text{-}POM}$ with alkenone $\delta^{13}C_{org}$ in the open ocean shows similar results as the averaged bins. Enrichment of bulk sediment $\delta^{13}C_{org}$, compared with $\delta^{13}C_{org\text{-}POM}$, noticeably increases towards the poles (Fig. 2A).

Atlantic planktonic $\delta^{13}C_{org\text{-}pforam}$ generally show similar values and trends to $\delta^{13}C_{org\text{-}POM}$ (Figs. 1 and 2). Our Northern North-Atlantic plankton tow JR271-B4 just south of Iceland shows a G. bulloides $\delta^{13}C_{org\text{-}pforam}$ value of −22.5‰ which is very similar to that

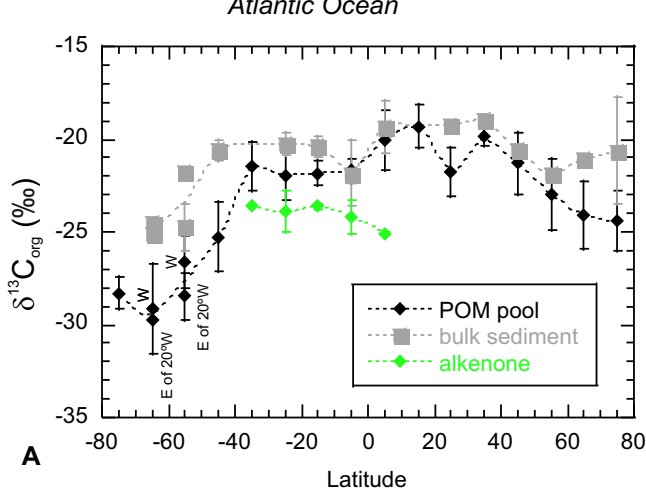

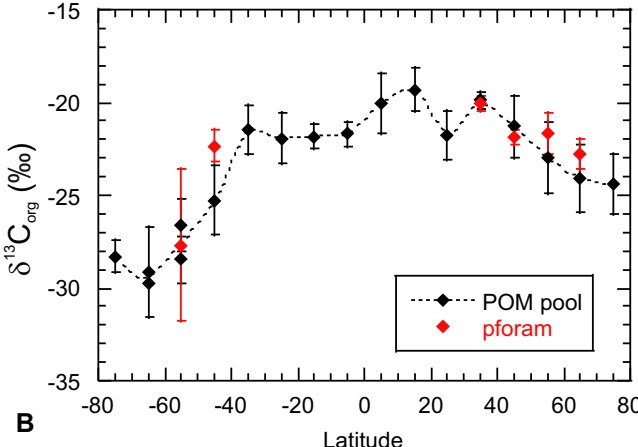

**Fig. 2 | Atlantic latitudinal transects. A** Averaged transects (10° latitude boxes) of $\delta^{13}C_{org\text{-}POM}$, $^{13}C_{org}$ of bulk sediments and alkenone $\delta^{13}C$. Black diamonds show organic carbon isotopes of particulate organic matter (POM), grey boxes $^{13}C_{org}$ of bulk sediments, and green diamonds alkenone $\delta^{13}C$. Means and standard deviations were calculated for sample sizes of 2 and above. On average, $n = 14$ for $\delta^{13}C_{org\text{-}POM}$, for $^{13}C_{org}$ of bulk sediments $n$ is 4 on average, and 6 for alkenone $\delta^{13}C$. Latitudes between 60–70° S and 50–60° S are split in an eastern and western section at 20° longitude for POM. **B** Averaged transects (10° latitude boxes) of $\delta^{13}C_{org\text{-}POM}$ and $\delta^{13}C_{org\text{-}pforam}$. Black diamonds show organic carbon isotopes of particulate organic matter (POM), whereas red diamonds show $\delta^{13}C_{org\text{-}pforam}$. Like in **A**, standard deviations are calculated for an $n = 14$ on average for $\delta^{13}C_{org\text{-}POM}$, and 3 for $\delta^{13}C_{org\text{-}pforam}$.

of the average $\delta^{13}C_{org\text{-}POM}$ (−22.3‰) in the area, whereas $\delta^{13}C_{org\text{-}pforam}$ at JR271-B5 (*T. quinqueloba* −22.0‰) in the Norwegian Sea appears enriched compared with the average $\delta^{13}C_{org\text{-}POM}$ (−25.9‰). Nearby RAPiD 6-3B and RAPiD 11-7B (62° N) core-tops *G. bulloides* $\delta^{13}C_{org\text{-}pforam}$ values of −22.9‰ and −23.8‰ are similar to average $\delta^{13}C_{org\text{-}POM}$ (−22.6‰). At BOFS 14K (slightly to the south at 58° N), planktonic foraminifera $\delta^{13}C_{org}$ (*G. inflata*, −20.9‰) is somewhat enriched compared with $\delta^{13}C_{org\text{-}POM}$ (average −22.6‰). At RAPID 39-28B (55° N, 52° W), *N. pachyderma* $\delta^{13}C_{org\text{-}pforam}$ is −22.5‰, but there is no nearby $\delta^{13}C_{org\text{-}POM}$ data to compare with.

At lower latitudes, $\delta^{13}C_{org\text{-}pforam}$ measurements are enriched compared to those from higher latitudes. At ODP Site 1308 (50° N), $\delta^{13}C_{org\text{-}pforam}$ (*G. inflata* −21.6‰, *G. hirsuta* −22.2‰) are very similar to the one nearby data point $\delta^{13}C_{org\text{-}POM}$ (−22.1‰). For ODP Site 1057 (32° N) there is no available nearby $\delta^{13}C_{org\text{-}POM}$ data. At this location, $\delta^{13}C_{org\text{-}pforam}$ (−19.6‰ to −20.5‰, excluding *T. sacculifer*) are somewhat

enriched when compared to $\delta^{13}C_{org\text{-}POM}$ (average −21.5‰) from the south (between 26–28° N and around 68° W), but similar to $\delta^{13}C_{org\text{-}POM}$ (average −19.8‰) from further northeast (between 40–45° N and 41–47° W). $\delta^{13}C_{org\text{-}pforam}$ of symbiont bearing *T. sacculifer* at ODP 1057 is enriched compared with *G. inflata*, *N. dutertrei* and *P. obliquiloculata* samples from the same Site, confirming that symbiont preferential use of isotopically light carbon causes enrichment in $\delta^{13}C_{org\text{-}pforam}$. We observe no differences between non-symbiont carrying and facultative symbiont carrying species (Table 2).

For the Southern Ocean, plankton net locations are located furthest to the south in the Scotia Sea, and show values between −25‰ (*G. bulloides*, 56.5° S) and −32.3‰ (*N. pachyderma*, 60° S), which agree well with nearby POM data (Fig. 1). Sedimentary $\delta^{13}C_{org\text{-}pforam}$ from ODP 1088 (41° S) varies between −21.0‰ (*G. inflata*) and −22.7‰ (*G. truncatulinoides*; *G. bulloides* is −21.7‰). At ODP 1090 (43° S), values are slightly more depleted, but show an inverse trend, with heavier values for *G. truncatulinoides* (−22.3‰) and more depleted values for *G. inflata* (−23.2‰, *G. bulloides* −23.1‰). $\delta^{13}C_{org\text{-}POM}$ is variable in the area; -21‰ just to the east, and between −22.5‰ and −27.1‰ to the south. While there are uncertainties relating to longer term and seasonal changes in $\delta^{13}C_{org}$ of POM, our new results suggest that planktonic foraminifera $\delta^{13}C_{org}$ are very similar to the $\delta^{13}C_{org}$ of POM (Figs. 1 and 2).

Our glacial *G. bulloides* sample has a $\delta^{13}C_{org\text{-}pforam}$ of −18.9‰ (Table 2).

## Discussion

Our plankton-net samples were collected in 2012 and 2013, whereas POM samples were collected between 1960 and 2010[42]. Young et al.[19] show that $\delta^{13}C_{org\text{-}POM}$ at the Ocean Flux Program (OFP) sediment trap off Bermuda has decreased by -1.5‰ between 1960 and 2010 (also see ref. 42), in relation to rising $CO_2$ and/or the Suess effect which leads to a decrease in $\delta^{13}C$ of DIC[51] and the POM subsequently produced from it[52]. If the recent decrease in $\delta^{13}C_{org\text{-}POM}$ changes is global (e.g. ref. 53) this would affect our POM versus planktonic foraminifera-bound $\delta^{13}C_{org}$ comparison; for example, $\delta^{13}C_{org\text{-}POM}$ samples predating planktonic foraminifera plankton-net samples from 2012 and 2013 are likely to be relatively enriched in $^{13}C$ due to a limited Suess effect. However, our Holocene core-top sediments are from relatively low accumulation rate areas which are unlikely to be influenced by the Suess effect (e.g. foraminifera specimens from the core-top samples predate POM sampling by centuries to millennia). Furthermore, it is currently unclear if there have been any underlying longer-term changes in $\delta^{13}C_{org\text{-}POM}$ through the Holocene. Because of these uncertainties, and uncertainties relating to short-term (seasonal) changes we have not attempted to correct our comparison of $\delta^{13}C_{org\text{-}pforam}$ with $\delta^{13}C_{org\text{-}POM}$ for such effects.

Culturing experiments by Uhle et al.[30] show $^{13}C$ depletion between planktonic foraminifera cytoplasm $\delta^{13}C_{org}$ and their food source; −3.5‰ for the symbiont-barren species *G. bulloides* and −2.4‰ for the symbiont-bearing *O. universa*. This is at odds with our results and other work (e.g. refs. 40, 54) which suggest there is no or relatively minor (-1‰) enrichments in $^{13}C$ from resource to consumer. The difference in $^{13}C$ enrichment between the culture study and our field observations probably relates to biosynthesis of the fatty and amino acids of shrimp fed cultured foraminifera[30]; such a shrimp-rich diet is likely not representative of the foraminifera's natural diet. It is also possible that the test-bound organic material in planktonic foraminifera may be enriched in $^{13}C$ compared with that of the cell cytoplasm. However, *G. bulloides* in plankton tow JR271 B4 (with cytoplasm) and nearby core top specimens from RAPiD 11-7B and 06-3B record very similar $\delta^{13}C_{org}$ values, which suggests that the cytoplasm $\delta^{13}C_{org}$ of planktonic foraminifera are similar to that held within the carbonate-bound organic lining. In addition, in the Sargasso Sea, planktonic foraminifera-bound organic matter nitrogen isotopes were found to be similar to that of POM[55].

Along our Atlantic transect, compared with other archives (e.g. bulk sedimentary $\delta^{13}C_{org}$ and $\delta^{13}C$ of alkenones), $\delta^{13}C_{org\text{-}pforam}$ fits best within the range of proximal $\delta^{13}C_{org\text{-}POM}$. The mean offset between bulk sedimentary $\delta^{13}C_{org}$ and $\delta^{13}C_{org\text{-}POM}$ is $-2.6 \pm 1.9$‰ ($n = 15$), whereas that of $\delta^{13}C_{org\text{-}pforam}$ and $\delta^{13}C_{org\text{-}POM}$ is $0.9 \pm 1.2$‰ ($n = 6$), reducing to $0.5 \pm 0.9$‰ ($n = 5$) if data between 40° and 50° S, with highly variable $\delta^{13}C_{org\text{-}POM}$, are excluded. A Student's $t$ test confirms that $\delta^{13}C_{org\text{-}POM}$ and $\delta^{13}C_{org\text{-}pforam}$ (including data from between 40° and 50° S) are statistically similar ($p = 0.13$), whereas $\delta^{13}C_{org\text{-}POM}$ and bulk $\delta^{13}C_{org}$ are statistically different ($p < 0.05$). This difference between $\delta^{13}C_{org\text{-}POM}$ and bulk $\delta^{13}C_{org}$ may arise from multiple potential issues affecting bulk $\delta^{13}C_{org}$: 1—diagenetic alteration, the effect of which typically increases with water depth and burial in sediment; 2—input of allochthonous organic carbon, including aeolian and riverine input of terrestrial plant material, 3—the nature, and associated differential preservation of detrital organic matter in sediments, and 4—redeposition of sediments transported from areas of erosion[27,56]. The similarity between $\delta^{13}C_{org\text{-}POM}$ and $\delta^{13}C_{org\text{-}pforam}$ agrees with previous work suggesting that trophic $\delta^{13}C_{org}$ fractionation is relatively minor[40,54], at least at the planktonic foraminifera level, supporting the proposition that $\delta^{13}C_{org\text{-}pforam}$ is a suitable proxy to reconstruct past upper ocean $\delta^{13}C_{org\text{-}POM}$.

There are three previous studies looking at $\delta^{13}C_{org\text{-}pforam}$, all aiming to use this method as a proxy to reconstruct atmospheric $CO_2$[28,34,39]. For the studies of Maslin et al.[39] and Swart et al.[28] we can compare $\delta^{13}C_{org\text{-}pforam}$ with local $\delta^{13}C_{org\text{-}POM}$. In both cases, $\delta^{13}C_{org\text{-}pforam}$ are depleted compared with local $\delta^{13}C_{org\text{-}POM}$. Maslin et al.[39], working in the Atlantic, also measured bulk sediment $\delta^{13}C_{org}$ which, while showing similar results to their $\delta^{13}C_{org\text{-}pforam}$, appear depleted by ~$-4.5$‰ compared with our core top compilation in Fig. 1. In their study Maslin et al.[39] used a dialysis step, and it is likely that this step removes labile organic material, potentially causing depletion of the remaining $\delta^{13}C_{org}$. The recent work of Swart et al.[28], in the Pacific, uses a different approach from Maslin et al.[39], with a different instrumental set-up and cleaning protocol. The $\delta^{13}C_{org\text{-}pforam}$ results of Swart et al.[28] are, however, also depleted ($-7$‰) compared with local $\delta^{13}C_{org\text{-}POM}$. Swart et al.[28] propose that this discrepancy is a result of the chemical composition of foraminifera bound organic material, attributing the depleted $\delta^{13}C_{org\text{-}pforam}$ results to a substantial lipid component. This is in contrast with previous studies suggesting that the foraminifera-bound organic material is predominantly composed of polysaccharides and proteins[29,36,57]. Comparable normalization standards were used in our and Swart et al.[28] studies, hence it is unlikely that instrumentation was responsible for the contrasting results. Instead, we assess whether different cleaning protocols applied prior to analysis could be responsible for the contrasting results.

In our approach we opted for a methodology that minimizes potential labile organic carbon loss, and potential contamination. Prior to analyses our samples were not exposed to temperatures exceeding 40–50 °C and we used acid vapour at room temperature to remove the calcium carbonate tests. We also minimized the introduction of non-laboratory grade/ and non-ultra-pure chemicals following the cleaning steps to limit potential sources of contamination. In their cleaning steps, Swart et al.[28] introduced two steps that exposed the foraminifera to elevated temperatures: 1—a reductive cleaning test where the sample vials were put in an 80 °C bath, and 2—an oxidative cleaning test where samples were autoclaved for 2 h at 122 °C. Autoclaving can cause hydrolysis and has been associated with changes in the chemical structure of soil organic matter and waste water[58,59]. While the foraminifera test may protect the organic carbon compounds bound within from aqueous cleaning methods, it cannot protect the organic carbon from extreme pressure and temperature changes. Heating the foraminifera to 80/122 °C may cause alteration of labile organic compounds bound to the carbonate lattice, potentially into smaller compounds and/or gaseous losses. Once the calcite test is removed, such gasses would be released. We hypothesize that the depleted $\delta^{13}C_{org\text{-}pforam}$ measured by Swart et al.[28] are an artefact of the sample treatment, with more labile organic carbon lost due to heating during their cleaning process. We suggest that only an approach that analyses all of the organic components held within foraminifera-bound organic carbon, as applied here, can produce comparable results to $\delta^{13}C_{org\text{-}POM}$.

One of the limitations of our approach is the large sample size needed to obtain meaningful analyses. To widen the proxy's application, sample sizes would need to be reduced, which may be facilitated by tailoring the instrumental set-up. It would be worthwhile assessing whether the differences between ours and the Swart et al.[28] cleaning protocols are predictable, in which case a derivative calibration may be developed to enable the analyses of samples that requires considerably less material.

To test the application of our proxy method on planktonic foraminifera samples that have undergone burial, we also measured $\delta^{13}C_{org\text{-}pforam}$ on a glacial Southern Ocean sample (G. bulloides sample from ODP Site 1088). While we do not have $\delta^{13}C_{org\text{-}POM}$ information for this time-interval, we can use ice core atmospheric $p$CO$_2$ concentrations and estimates of G. bulloides calcification DIC (from inorganic carbon isotopes) and temperature (using Mg/Ca) to predict phytoplankton (coccolith) $\delta^{13}C_{org\text{-}POM}$, using the formulae of Hernandez-Almeida et al.[60]. To calculate calcification temperature, we used the calibration equation of Vazquez-Riveiros et al.[61] specifically developed for high latitude G. bulloides. With a calcification temperature of 6.6 °C (Mg/Ca = 1.2 mmol/mol), G. bulloides $\delta^{13}C_{calcite}$ of 0.46‰, and atmospheric $CO_2$ concentration of 192 ppmv, we estimate that glacial $\delta^{13}C_{org\text{-}POM}$ is $-18.6 \pm 0.7$‰, taking into account uncertainties relating to temperature estimates ($\pm 1$ °C, equal to $\pm 0.6$‰), $\delta^{13}C_{DIC}$ measurement ($\pm 0.1$‰), and ice core $CO_2$ ($\pm 10$ ppmv, equal to 0.3‰). This predicted value is very close to our actual $\delta^{13}C_{org\text{-}pforam}$ measurement of $-18.9$‰, illustrating that our method can be used to obtain reliable estimates of past $\delta^{13}C_{org\text{-}POM}$.

Our current knowledge of how the carbon isotopic composition of upper ocean POM has changed with time is extremely limited, yet the information gained from such insights is important to understand the global carbon cycle, especially on relatively long-time-scales. Measurements of down-core $\delta^{13}C_{org\text{-}pforam}$ will help us to understand the extent and magnitude of $\delta^{13}C_{org\text{-}POM}$ changes in the ocean in order to understand its implications on $\delta^{13}C$ of DIC at various timescales. This includes inferences relating to the ocean, terrestrial and atmospheric sources and sinks of $CO_2$. Since the Palaeocene, Earth's average temperature and $p$CO$_2$ have decreased, permanent ice-caps developed in the Southern and Northern Hemisphere, while ocean gateways opened and closed. The effects of these changes on $\delta^{13}C_{org\text{-}POM}$ and subsequently $\delta^{13}C_{DIC}$ remain unexplored. Kast et al.[62] recently reported the use of planktonic foraminifera organic carbon test-bound $\delta^{15}N$ to explore suboxia during the early Cenozoic between 70 and 25 million years ago. As nitrogen and carbon isotopes are extracted from the same medium (i.e. test-bound organic matter), we anticipate that planktonic foraminifera test-bound $\delta^{13}C_{org}$ can be applied over similar time windows to understand past changes in upper ocean $\delta^{13}C_{org\text{-}POM}$ as well as effects on the $\delta^{13}C$ of DIC in seawater.

## Methods

The $\delta^{13}C_{org}$ was measured on single-species planktonic foraminifera samples freshly picked from plankton net samples and from recent surface sediment samples (Supplementary Figs. 1 and 2 and Tables 1 and 2 provide age details in the Supplementary information file), all from the Atlantic Ocean (Table 1). Species analysed include *Globigerina bulloides*, *Globorotalia hirsuta*, *G. inflata*, *G. truncatulinoides*, *Neogloboquadrina pachyderma*, *N. dutertrei*, *Globigerinoides/Trilobus sacculifer*, *Pulleniatina obliquiloculata* and *Turborotalita quinqueloba*. *Trilobus sacculifer*, an upper ocean mixed layer species, is the only species that bears algal symbionts (e.g. *chrysophytes, dinoflagellates*); *N. dutertrei, G. inflata* and *P. obliquiloculata* are facultative symbiont-bearers[29,63].

Vertical plankton net hauls (200 m to surface, 100 μm mesh size) were collected from high latitudes (north and south Atlantic) during two research cruises (JR271 in 2012 and JR274 in 2013) as part of the UK Ocean Acidification Research Program (www. oceanacidification.org.uk). Plankton net samples were washed from the net into a bucket, and foraminifera allowed to settle under gravity. Foraminifera were collected from the bottom of the bucket using a hand pipette. Foraminifera were then washed over a 100 μm mesh with pH-adjusted Milli-Q (pH>8), oven dried (40–50 °C, 8–12 h) and stored in clean glass vials. For net-caught samples, planktonic foraminifera specimens with cytoplasm were analysed. No additional cleaning steps were applied prior to $\delta^{13}C_{org\text{-}pforam}$ analyses of net-caught planktonic foraminifera.

Core-top and surface sediment samples are from subtropical to polar latitudes (32–62° N, 41–42° S). Sediment samples were washed over a 63 μm sieve using demineralized water, and subsequently oven-dried at 40 °C. Specimens were picked from the dry-sieved >250 μm fraction. Typically, for organic carbon isotope analyses, ~15–25 μg of organic material was needed, which amounted to a minimum total mass of 45 mg $CaCO_3$, equivalent to between 1500 and 3500 specimens. There are no measurements from tropical latitudes, as it was not possible to pick sufficient mono-specific sample material from these highly diverse samples. Prior to analyses, foraminifera picked from the core top/surface sediments were crushed and cleaned following methods developed by Barker et al.[64] and Ren et al.[38] to remove adhered particles and sedimentary material within the test. The cleaning started with crushing of foraminifera between two glass slides to open up the chambers. Crushed samples were transferred to clean glass vials covered with aluminium foil and oven dried at 40 °C. All glassware used from this point on was cleaned with Teepol™ and combusted at 400 °C to remove any potential organic residues. Combusted aluminium foil was used to cover samples and line the lids of the glass vials. Dried samples were transferred into isotope vials using Pasteur pipettes and Milli-Q water. After topping up with Milli-Q water (to ~1 ml) samples were ultrasonicated to release clay and other fine material from the test walls. Following ultrasonification, samples were allowed to settle for 30 s after which the supernatant was pipetted off. This step was repeated several times until there was no more release of fine material. Samples were then soaked in 10 ml of 13% NaOCl for 4 h and agitated every half hour, to remove adhered organic material[40]. The NaOCl was pipetted off and the samples were rinsed with Milli-Q water six times, with an ultrasonic treatment at the last rinse to ensure all bleach was removed. Finally, samples were transferred to clean glass vials and oven dried at 40 °C. The same technique was applied to a glacial G. bulloides sample from ODP Site 1088.

$\delta^{13}C_{org}$ was measured on decalcified bulk sediments, mono-specific plankton tow material and core tops, and the one glacial planktonic foraminifera sample. The inorganic carbon fraction of bulk sediments, plankton tow foraminifera, and cleaned foraminifera was removed by 48 hour hydrochloric acid vapour treatment. Samples that continued to have bubble formation (indication that the HCl was still reacting with $CaCO_3$), were left for another 24 hours. Every 24 hours samples were dried at 40 °C, and the acid changed. An acid vapour treatment was preferred over dissolution in hydrochloric acid, as decanting of the HCl solution following decarbonation removes the soluble organic fractions. Ren et al. (2009) showed that cleaned and acidified planktonic foraminifera contained about 0.006% N. Assuming Redfield ratios, this would amount to about 0.03% C. Organic carbon contents of cleaned acidified planktonic foraminifera samples from core tops and the LGM are typically 0.03%, confirming that all non-bound organic C was oxidized.

Samples were weighed into tin capsules and combusted using a Eurovector elemental analyser. Resultant $CO_2$ from combustion was analysed for $\delta^{13}C$ using a Micromass Isoprime IRMS. The standard deviation of QC samples for $\delta^{13}C_{org}$ did not exceed 0.16‰. The QC sample is an in-house soil reference, calibrated to NIST standards sucrose anu (NIST 8542) and USGS40 (NIST 8573) as normalization standards. The standard deviation of duplicate planktonic foraminifera samples for $\delta^{13}C_{org}$ did not exceed 0.90‰.

## Data availability

Source data are provided with this paper.

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

## Acknowledgements

The work was supported by a UK Natural Environment Research Council (NERC) grant NE/K00087X/1, a UKRI Future Leaders Grant MR/S034293/1, and a Philip Leverhulme Prize (to BAAH). Thanks to Simon Chenery and Elliot Hamilton for analysing the Mg/Ca of the glacial ODP 1088 sample. We are grateful to Nick McCave for making sample material from RAPiD and BOFS14K available. This research used samples and/or data provided by the Ocean Drilling Program (ODP). ODP is sponsored by the U.S. National Science Foundation and participating countries (Natural Environment Research Council in UK) under management of Joint Oceanographic Institutions (JOI), Inc. The UK Ocean Acidification research programme was funded by the Department for Environment, Food and Rural Affairs, the NERC and the Department of Energy and Climate Change (NE/H017267/1, NE/H017097/1).

## Author contributions

B.A.A.H. conceived and coordinated the work. C.A., V.L.P., B.A.A.H. and T.P. picked all the sample materials. Cleaning of foraminifera was carried out by C.A., B.A.A.H. and T.P. C.A., S.B., T.P. and C.M. decalcified the samples, while H.G., A.S. and P.K. carried out the data analyses. B.A.A.H. constructed the figures and wrote the paper, with contributions from C.A., T.P., A.S., H.G., P.K., C.M., S.B., E.L.M., R.R., A.P. and V.L.P.

## Competing interests

The authors declare no competing interests.
