## [Peer Review File · Nature Communications]

Planktonic foraminifera organic carbon isotopes as archives of upper ocean carbon cyclingReviewers' comments:

Reviewer #1 (Remarks to the Author):

Review of "Planktonic foraminifera organic carbon isotopes as novel archives of upper ocean carbon cycling" by Hoogakker et al., submitted to Nature Communications, by Jesse Farmer

Hoogakker et al. present the first (to my knowledge) tow and core-top compilation of carbon isotopes within the organic matrix of planktonic foraminiferal tests. The authors find that the carbon isotopic composition of the planktonic foraminifera organic matrix is closely related to that of in situ particulate organic carbon. These results open the door for a potential $\delta^{13}\text{C}_{\text{POM}}$ proxy, which would be of great value for better understanding past changes in the global carbon cycle.

Although the number of samples are limited and the comparisons in this text are rather generalized to the basin-scale, the authors have put a tremendous amount of work into these samples, and the manuscript overall reads well. I have two major comments for the authors to consider in a revision, below, which I believe will strengthen the final manuscript. I am receptive to seeing this published following a round of revisions.

Comment 1 (L193-198 & Figure 2). I would like to know more about the compilation process, particularly for the South Atlantic. The process of binning by latitude greatly simplifies this story (and I commend the authors for doing so). At the same time, the binning may be particularly biased toward areas of high data density (such as coastal regions between $\sim 0^\circ$ and $\sim 30^\circ\text{S}$). How do you evaluate this bias? Furthermore, given that the $\delta^{13}\text{C}_{\text{org-pforam}}$ data come largely from the open basins, have you explored ways to more closely match these data with representative $\delta^{13}\text{C}_{\text{org-POM}}$ measurements? Does this at all change the comparison?

Comment 2 (L276-288). This is not a typical request of discussion sections, but I believe this paper should comment on the future suitability of the presented method. For comparison, it has taken foraminifera-bound N isotope measurements nearly a decade to become semi-routine, yet we only require 5 mg of foraminifera (despite there being less N in foraminifera-bound organic matter than C). From my perspective, I see issues with scaling up with approach insofar as 1500-3500 monospecific foraminifera are needed for each analysis. What are the prospects for reducing the sample requirements, and what future developments might aid in this regard?

Minor comments/edits

L27. Delete extra comma

L32. Add comma after "open ocean"

L37-41. To me the biggest potential impact of these are how they might change atmospheric CO_2 levels

L57. Specifically, glacial sea level low stands

L113. Error in reference formatting

L126-129. I would agree that this approach could be subject to fewer uncertainties, but describing the picking of 1500+ monospecific foraminifera specimens as "rapid" or "efficient" seems off the mark. Rephrase this.

L177. In the statement "this removes the soluble organic fractions", does "this" refer to dissolution in HCl or the HCl vapour treatment? Please specify.

L202-203 (& again on L211, L268-270). Recommend reporting all values to one decimal place for consistency with the majority of results and given the high standard deviations for $\delta^{13}\text{C}_{\text{org}}$ ($\leq 0.90\text{‰}$, L181).

L216-219. This comparison is important, as Maslin has the only other test-bound $\delta^{13}\text{C}$ measurements in foraminifera that I am aware of. Is it reasonable to think that the soluble organic fractions would be enriched in ^{13}C relative to the insoluble component (as this comparison would imply)? I imagine some work has been performed on this in the C isotope community that could be referenced here.

L246-250. It may help your point that N isotope work has noted little difference between the cytoplasm $\delta^{15}\text{N}$ and shell-bound $\delta^{15}\text{N}$ of net tow samples, although coretops are slightly isotopically elevated compared to net tows (Smart et al., 2018 GCA).

Reference to include: "Molecular and isotopic composition of foraminiferal organic linings" by Ní Fhlaithearta et al., Marine Micropaleontology (2013)

L499. Change "is" to "are"

Fig 1. I would suggest changing the color scheme to accentuate the planktonic foraminifera-bound $\delta^{13}\text{C}$ data. Currently the POM and alkenone data are most visible on this plot, while the foram data are rather hidden.

Reviewer #2 (Remarks to the Author):

This paper seeks to develop $\delta^{13}\text{C}$ of the organic linings of foraminifera as a proxy for the $\delta^{13}\text{C}$ of the particulate organic matter pool in the past oceans. To do this, the authors combine tow and sediment trap data to investigate how well foraminifera cytoplasm/linings track the reservoir in the open ocean; this is because as foraminifera consume POC, their own organic matter should reflect $\delta^{13}\text{C}$ -POC.

While this is a compelling proxy proposal, I do not believe that the paper as it stands lays out a convincing argument that this can now be robustly applied to the past. The main reason for this is that the paper does not include any measurements of $\delta^{13}\text{C}$ from fossil foraminifera. Firstly, tow foraminifera will include cytoplasm, as the authors note. This is interesting from a proxy development and process standpoint, but it has no relevance for paleo-studies as the cytoplasm is converted into gametes. Secondly, it is not clear if the primary $\delta^{13}\text{C}$ signal is preserved in foraminifera linings that have undergone burial. While the outlook is promising given the success of the $\delta^{15}\text{N}$ proxy, it is not clear that beyond coretops, sedimentary foraminifera will prove a useful archive.

Other issues in the manuscript arise from the justification for developing this proxy as well as the treatment of the results. Firstly, it is not clear to a general reader exactly why understanding the $\delta^{13}\text{C}$ of the POC pool will help to illuminate past carbon cycle dynamics and important climate information. See specific comments on this below.

I also believe that there is room here for a much more detailed investigation of what the data show. For example, the authors have measured many different species, ranging from symbiont-bearing to barren, yet there is no discussion of their distinct $\delta^{13}\text{C}$ signatures (or lack thereof). E.g., if the $\delta^{13}\text{C}$ of the foraminiferal microenvironment is increased due to symbiont photosynthesis, will this affect the $\delta^{13}\text{C}$ of the organic lining? If the proxy is to be applied to the past, investigators need to know if species-specific effects are to be expected.

Overall, I agree with the authors that this shows promise as a system but further study is needed to support the main conclusion of the study that the method will be useful for fossil reconstructions.

I hope these line by line comments will be useful to the authors.

Abstract: Define $\delta^{13}\text{C}$
Line 27: delete comma

Intro paragraph: Here, for the more general reader, it is difficult to know what constraining the $\delta^{13}\text{C}$ values of these different reservoirs will tell us about long-term carbon burial, etc.

Start with why we need to know the $\delta^{13}\text{C}$ of DIC. This is used as the justification in many places for constraining $\delta^{13}\text{C}$ -DOM; What can $\delta^{13}\text{C}$ of DIC tell us about the carbon cycle? Then, why do we need to know $\delta^{13}\text{C}$ of POM? I.e., why is it problematic that $\delta^{13}\text{C}$ -bulk C_{org} is insufficient? What, specifically, will the $\delta^{13}\text{C}$ of POM tell us about these processes?

The background on foraminifera is not directly related to the topic at hand. Tie this background info back to the overall motivation of the study. E.g., why do we need to know about about foraminiferal gametogenesis in order to understand the $\delta^{13}\text{C}$ of their organic linings? Why is it important for this study that foraminiferal flux often occurs in pulses/blooms?

51-53: How will knowing the actual $\delta^{13}\text{C}$ -POM help us to further understand SSTs and air-sea fractionation? i.e. what will we learn about the climate from this finding?

107: The mention of pervasive organic matter diagenesis without further investigation makes it unclear whether this will be an issue for applying the proxy to fossil foraminifera.

148: Clarify that these analyses include cytoplasm, and state explicitly

155: specify 45 mg CaCO_3

169: Explain the purpose of NaOCl soaking

190: What is the variability over this long period (1960-2010); giving us insight into whether this is a robust comparison?

205: Compare tow with measured first; then coretops with measured, for a clearer analysis.

220: more enriched than $\delta^{13}\text{C}$ -POM? Please specify.

Response to reviewers comments:

We are grateful for the two thoughtful and constructive reviews for our manuscript. We have revised the manuscript following the reviewers' suggestions.

In the text that follows we address the reviewers' comments, with the comments in black and our response in green. After the response to reviewers comments who show the revised text and Figure 1.

Reviewer #1 (Remarks to the Author):

Review of "Planktonic foraminifera organic carbon isotopes as novel archives of upper ocean carbon cycling" by Hoogakker et al., submitted to Nature Communications, by Jesse Farmer

Hoogakker et al. present the first (to my knowledge) tow and core-top compilation of carbon isotopes within the organic matrix of planktonic foraminiferal tests. The authors find that the carbon isotopic composition of the planktonic foraminifera organic matrix is closely related to that of in situ particulate organic carbon. These results open the door for a potential $\delta^{13}\text{C}_{\text{POM}}$ proxy, which would be of great value for better understanding past changes in the global carbon cycle.

Although the number of samples are limited and the comparisons in this text are rather generalized to the basin-scale, the authors have put a tremendous amount of work into these samples, and the manuscript overall reads well. I have two major comments for the authors to consider in a revision, below, which I believe will strengthen the final manuscript. I am receptive to seeing this published following a round of revisions.

Comment 1 (L193-198 & Figure 2). I would like to know more about the compilation process, particularly for the South Atlantic. The process of binning by latitude greatly simplifies this story (and I commend the authors for doing so). At the same time, the binning may be particularly biased toward areas of high data density (such as coastal regions between $\sim 0^\circ$ and $\sim 30^\circ\text{S}$). How do you evaluate this bias? Furthermore, given that the $\delta^{13}\text{C}_{\text{org-pforam}}$ data come largely from the open basins, have you explored ways to more closely match these data with representative $\delta^{13}\text{C}_{\text{org-POM}}$ measurements? Does this at all change the comparison?

The reviewer is correct that there could be a potential bias in the South Atlantic when comparing carbon isotopes of POM, sediments and alkenones, especially when it comes to bulk sediments, which have a high sample density along the west African margin where there is no POM data. If we focus our comparison of POM versus sediments (note there is only one datapoint per bin here!) and alkenones on the open South Atlantic between 10 degrees North and 30 degrees South, then the difference between POM and sediments is actually increased, on average by 1 per mil. This is while both open ocean POM and sediment $\delta^{13}\text{C}_{\text{org}}$ are on average enriched compared with continental margins. Comparison of POM with alkenone $\delta^{13}\text{C}_{\text{org}}$ in the open ocean shows similar results as the averaged bins. This information has been added to our results

With regards to the $\delta^{13}\text{C}_{\text{org-pforam}}$ data, in the text of the results section (lines -206-237) we compare these data with localized $\delta^{13}\text{C}_{\text{org-POM}}$ measurements where possible. This more regional comparison agrees with the findings in Figure 2B.

Comment 2 (L276-288). This is not a typical request of discussion sections, but I believe this paper should comment on the future suitability of the presented method. For comparison, it has taken foram-bound N isotope measurements nearly a decade to become semi-routine, yet we only require 5 mg of foraminifera (despite there being less N in foram-bound organic matter than C). From my perspective, I see issues with scaling up with approach insofar as 1500-3500 monospecific foraminifera are needed for each analysis. What are the prospects for reducing the sample requirements, and what future developments might aid in this regard?

This is a valid point raised by the reviewer. Reconstructions using this proxy are currently limited by the large samples size needed. In the future we plan to look at reducing sample sizes facilitated by changes in instrument set-up.

Importantly though, our approach needing large samples sizes, is the first to show that there is minimal fractionation between planktonic foraminifera and their diet. All other previous attempts with methods that need much less material have reported values that are largely depleted compared with $\delta^{13}\text{C}_{\text{org-POM}}$. For example, results of Maslin et al. 42, using only 50 specimens of planktonic foraminifera, are depleted by 3 to 4‰. Maslin et al. 42 used a dialyses step, which would have removed labile organic material. In a more recent study, Swart et al (2021) analyzed organic carbon isotopes on tropical Pacific core tops, needing only 20 nM of organic carbon, a thousand times less than needed in our study. In Swart's analyses foraminifera samples were heavily depleted (-7 per mil) compared with $\delta^{13}\text{C}_{\text{org-POM}}$, which they related to a substantial lipid contribution to the foraminifera organic material. However, other studies (Stott et al., 1992, Ní Fhlaithearta et al., 2013; Schiebel and Hemleben, 2017) claim that foraminifera organic material is mainly composed of polysaccharides and proteins. Swart et al. (2021) carried out an oxidative cleaning step by autoclaving the samples. Autoclaving of samples is known to cause hydrolysis of organic material. The depleted results found by Swart et al (2021) supports the inference that hydrolysable materials (carbohydrates, amino acids) are enriched in ^{13}C compared with lipids and non-hydrolysable materials (Hwang and Druffel, 2003). We have added a paragraph to our discussion.

Minor comments/edits

L27. Delete extra comma

Done

L32. Add comma after "open ocean"

Done

L37-41. To me the biggest potential impact of these are how they might change atmospheric CO2 levels

Absolutely, added CO2 to this sentence.

L57. Specifically, glacial sea level low stands

Done

L113. Error in reference formatting

Done

L126-129. I would agree that this approach could be subject to fewer uncertainties, but describing the picking of 1500+ monospecific foraminifera specimens as “rapid” or “efficient” seems off the mark. Rephrase this.

We have rephrased ‘more rapid and efficient’ to better.

L177. In the statement “this removes the soluble organic fractions”, does “this” refer to dissolution in HCl or the HCl vapour treatment? Please specify.

This sentence has been changed to “An acid vapour treatment was preferred over dissolution in hydrochloric acid, as decanting of the HCl solution following decarbonation removes the soluble organic fractions.”

L202-203 (& again on L211, L268-270). Recommend reporting all values to one decimal place for consistency with the majority of results and given the high standard deviations for d13Corg ($\leq 0.90\%$, L181).

We have changed the reported values to one decimal point. As the standard deviations for foraminifera is already provided in the methods we didn’t think it worthwhile to repeat this for all the foraminifera samples.

L216-219. This comparison is important, as Maslin has the only other test-bound d13C measurements in foraminifera that I am aware of. Is it reasonable to think that the soluble organic fractions would be enriched in 13C relative to the insoluble component (as this comparison would imply)? I imagine some work has been performed on this in the C isotope community that could be referenced here.

We have now moved this comparison to the discussion part as recently another study was published looking at core tops from the tropical eastern Pacific, which also showed very depleted values (see response to your comment 2). There has been various work carried out on bulk sediments. This however holds at least 30 to 50 times more organic material than that held within foraminifera calcite, and so the results are dominated by more recalcitrant organic carbon held within these sediments, and cannot provide much insight here.

L246-250. It may help your point that N isotope work has noted little difference between the cytoplasm d15N and shell-bound d15N of net tow samples, although coretops are slightly isotopically elevated compared to net tows (Smart et al., 2018 GCA).

We thank the reviewer for this. We note that the shell standard deviation would suggest that statistically there is no difference, and have added a sentence at the end of this paragraph ‘For planktonic foraminifera-bound organic matter nitrogen isotopes Smart et al. (2018) also observed no difference in the Sargasso Sea.’

Reference to include: “Molecular and isotopic composition of foraminiferal organic linings” by Ní Fhlaithearta et al., Marine Micropaleontology (2013)

We added the following sentence: “In a culturing study, Ní Fhlaithearta et al (2013) found that metabolic carbon is the main carbon source fixed within benthic foraminifera organic linings.”

L499. Change “is” to “are”

Done.

Fig 1. I would suggest changing the color scheme to accentuate the planktonic foraminifera-bound d13C data. Currently the POM and alkenone data are most visible on this plot, while the foram data are rather hidden.

We have done this, colours for new data (foram POM and bulk sediments) are in bigger format, and the symbols for the forams are in brighter colours.

Reviewer #2 (Remarks to the Author):

This paper seeks to develop d13C of the organic linings of foraminifera as a proxy for the d13C of the particulate organic matter pool in the past oceans. To do this, the authors combine tow and sediment trap data to investigate how well foraminifera cytoplasm/linings track the reservoir in the open ocean; this is because as foraminifera consume POC, their own organic matter should reflect d13C-POC.

We use planktonic foraminifera-bound organic carbon isotopes, where foraminifera were obtained from plankton tow and sediment cores, and compare this with $\delta^{13}\text{C}_{\text{org-POM}}$.

Planktonic foraminifera from sediment traps were not used, as the chemical used to preserve material caught in sediment traps, e.g. formalin, causes considerable contamination in organic carbon isotopes.

While this is a compelling proxy proposal, I do not believe that the paper as it stands lays out a convincing argument that this can now be robustly applied to the past. The main reason for this is that the paper does not include any measurements of d13C from fossil foraminifera. Firstly, tow foraminifera will include cytoplasm, as the authors note. This is interesting from a proxy development and process standpoint, but it has no relevance for paleo-studies as the cytoplasm is converted into gametes. Secondly, it is not clear if the primary d13C signal is preserved in foraminifera linings that have undergone burial. While the outlook is promising given the success of the d15N proxy, it is not clear that beyond coretops, sedimentary foraminifera will prove a useful archive.

We thank the reviewer for this comment. In our revised manuscript we now include the analysis of a foraminifera-bound organic carbon isotope sample from the last glacial period to demonstrate the application on samples that have undergone burial. This sample's $\delta^{13}\text{C}_{\text{org-pforam}}$ is as would be expected for this location, under lower glacial seawater temperatures and atmospheric CO₂ values. We have added text to the discussion section of the manuscript.

Other issues in the manuscript arise from the justification for developing this proxy as well as the treatment of the results. Firstly, it is not clear to a general reader exactly why understanding the d13C of the POC pool will help to illuminate past carbon cycle dynamics and important climate information. See specific comments on this below.

We thank the reviewer for highlighting this issue. We have added more text to the introductory paragraph to provide more justification for the development of the proxy.

I also believe that there is room here for a much more detailed investigation of what the data show. For example, the authors have measured many different species, ranging from symbiont-bearing to barren, yet there is no discussion of their distinct $\delta^{13}\text{C}$ signatures (or lack thereof). E.g., if the $\delta^{13}\text{C}$ of the foraminiferal microenvironment is increased due to symbiont photosynthesis, will this affect the $\delta^{13}\text{C}$ of the organic lining? If the proxy is to be applied to the past, investigators need to know if species-specific effects are to be expected.

We have added details in the results section about this. We don't observe a difference between facultative and non-symbiont bearer species. Our analyses for a symbiont-bearing species however does show enrichment, which confirms previous culturing work (e.g. Uhle et al. 1997).

Overall, I agree with the authors that this shows promise as a system but further study is needed to support the main conclusion of the study that the method will be useful for fossil reconstructions.

I hope these line by line comments will be useful to the authors.

Abstract: Define $\delta^{13}\text{C}$

Done.

Line 27: delete comma

Done.

Intro paragraph: Here, for the more general reader, it is difficult to know what constraining the $\delta^{13}\text{C}$ values of these different reservoirs will tell us about long-term carbon burial, etc.

Start with why we need to know the $\delta^{13}\text{C}$ of DIC. This is used as the justification in many places for constraining $\delta^{13}\text{C}$ -DOM; What can $\delta^{13}\text{C}$ of DIC tell us about the carbon cycle? Then, why do we need to know $\delta^{13}\text{C}$ of POM? I.e., why is it problematic that $\delta^{13}\text{C}$ -bulk C_{org} is insufficient? What, specifically, will the $\delta^{13}\text{C}$ of POM tell us about these processes?

The background on foraminifera is not directly related to the topic at hand. Tie this background info back to the overall motivation of the study. E.g., why do we need to know about about foraminiferal gametogenesis in order to understand the $\delta^{13}\text{C}$ of their organic linings? Why is it important for this study that foraminiferal flux often occurs in pulses/blooms?

We have followed the reviewers advice and made various changes to the introduction and foraminifera background, only discussing information that directly matters to the proxy.

51-53: How will knowing the actual $\delta^{13}\text{C}_{\text{POM}}$ help us to further understand SSTs and air-sea fractionation? i.e. what will we learn about the climate from this finding?

Upon rereading this sentence we realised that the $\delta^{13}\text{C}_{\text{DIC}}$ may be out of place, and we have deleted this from the line, as the focus in this particular paragraph is on reconciling $\delta^{13}\text{C}_{\text{org-POM}}$.

Not sure we understand the reviewers point. We are using $\delta^{13}\text{C}_{\text{org-POM}}$ to learn what the *effects* of SST and air to sea fractionation are on this. We are not claiming that there is more to find out about climate.

Our revised text (between “ ”):

“To illustrate the complexity of reconciling $\delta^{13}\text{C}_{\text{org-POM}}$: during ice ages/ glacial stages, lower atmospheric CO_2 concentrations and colder sea surface temperatures are thought to have caused a reduction in air-sea fractionation, which in turn would have caused enrichment in $\delta^{13}\text{C}_{\text{org-POM}}$ ¹². Bulk sediment $\delta^{13}\text{C}_{\text{org}}$ measurements from the Atlantic over the last glacial-interglacial cycle however do not show uniform enrichment, indicating that processes other than $p\text{CO}_2$ /temperature can overprint the signal in bulk sediments¹²⁻¹⁶.” We have moved the discussion that follows about bulk sediments to the discussion and then finish this paragraph with “Here we assess whether planktonic foraminifera-bound $\delta^{13}\text{C}_{\text{org}}$ ($\delta^{13}\text{C}_{\text{org-pforam}}$) provides a better representation of suspended $\delta^{13}\text{C}_{\text{org-POM}}$ than $\delta^{13}\text{C}_{\text{org}}$ of bulk sediments. We focus on a N-S Atlantic Ocean transect, using living and recent (Holocene) planktonic foraminifera.”

107: The mention of pervasive organic matter diagenesis without further investigation makes it unclear whether this will be an issue for applying the proxy to fossil foraminifera.

This could indeed potentially provide issues. However, if diagenesis is an issue, this should also involve planktonic foraminifera bound nitrogen isotopes, as these measurements are on the same organic material. Kast et al. (2019) recently successfully applied this approach back to ~70 Ma. For our revised manuscript, we have now analyzed a sample (Southern Ocean) from the last glacial period, from around 20,000 years ago to illustrate the potential for applications in the past.

148: Clarify that these analyses include cytoplasm, and state explicitly

We have adjusted the sentence to reflect this.

155: specify 45 mg CaCO_3

We have added this detail in the revised text.

169: Explain the purpose of NaOCl soaking

We have added this to the revised text.

190: What is the variability over this long period (1960-2010); giving us insight into whether this is a robust comparison?

Unfortunately, apart from the work by Young et al. (2013) there are no repeat measurements for the various locations and it is not possible to assess this variability.

205: Compare tow with measured first; then core tops with measured, for a clearer analysis.

We have split this discussion in a regional comparison, starting with the North Atlantic, working our way to the South Atlantic. For the North Atlantic part we do start with tow measurements, followed

by core-tops, but it is the other way around for the South Atlantic. In our revised manuscript we have now switched this, that we first discuss Southern Ocean plankton tow data, followed by the core tops.

220: more enriched than d13C-POM? Please specify.

Compared with those of higher latitudes. We have added this detail in the revised text.

REVIEWER COMMENTS

Reviewer #1 (Remarks to the Author):

R1 review of Hoogaaker et al., "Planktonic foraminifera organic carbon isotopes as novel archives of upper ocean carbon cycling," submitted to Nature Communications, by Jesse Farmer

I am torn on the revised version of this manuscript. On one hand, the authors' fascinating results remain in place (they have even added one new analysis), and the fundamental motivation and framing of this study remains solid.

On the other hand, there are two major deficiencies in the submitted manuscript. First, since the initial submission, the study of Swart et al. (2021, GCA) was published investigating the $\delta^{13}\text{C}$ of planktonic foraminifera-bound organic matter in the eastern tropical Pacific. I do not believe the existence of the Swart et al. publication by itself fundamentally changes the novelty or significance of this current study. However, it is critical that the authors' compare to the methods and results of Swart et al. throughout for completeness. As it stands, the authors only mention this study toward the end of the manuscript, and here largely to raise a point about methodological differences. This is not sufficient.

Second, there are numerous errors in the submitted manuscript that made for a challenging review. There was no Figure 2 in the uploaded manuscript, nor a tracked changes version. However, at least some of the additions mentioned by the authors in the response to reviews are not well integrated with the existing manuscript. For instance, there is no presentation of the new LGM data within the Results section or on any figure (I should note that these does appear later in the discussion). And again, on L269-271, the observation of no difference between cytoplasm and foraminifera-bound organic matter $\delta^{15}\text{N}$ is thrown on the end of this paragraph; it is not clear to the reader how this observation supports the authors' reasoning.

Ultimately, the authors' may be given another opportunity for major revisions to better integrate the new literature on foraminifera-bound organic material $\delta^{13}\text{C}$ with their results and improve the readability and presentation of the manuscript. A cautionary note is that these changes may lead to a longer and more field-specific manuscript that would be better suited to a different journal, but I leave that decision to the editors.

Line-by-line comments.

L47. Suggest change to "the $\delta^{13}\text{C}$ of specific benthic foraminifera".

L53-56. It would greatly help this paragraph if this proxy-specific point were removed, as this would keep the paragraph focused on big-picture concerns.

L58-63. This point could be more strongly made with an actual illustration (a figure).

L68-86 This section "Distribution of suspended $\delta^{13}\text{C}_{\text{org-POM}}$ in the Atlantic Ocean" seems out of place and a mash of introduction, results, discussion, and interpretation. I'm not quite sure what to do with this, but perhaps saving it for after the planktonic foraminifera organic carbon and materials & methods sections would help.

L80. Add hyphen "13C-depleted"

L93: move semi-colon to after "omnivorous": "Planktonic foraminifera are omnivorous; their diet varies between species and habitat, but includes..."

L120. Swart et al. should also be cited here.

L143. Add comma after first "species"

L152. Check this sentence for typo or grammatical error.

L154. Add hyphen, e.g. "net-caught"

L155. Recommend specifying latitude range here e.g., 32 to 62 °N, 41 to 42 °S

L174. What volume of bleach was used? Was any evaluation done to see if there was sufficient oxidative capacity in the bleach to oxidize all non-bound organic C? For context, Ren et al. used 10 mL of 13% bleach for 6 hours on up to 10 mg of foraminifera.

L183. HCl

L188-190. What were the quality control samples? And are the "duplicate samples for $\delta^{13}\text{C}_{\text{org}}$ " referring to $\delta^{13}\text{C}_{\text{org-pforam}}$? Please specify.

L203. There is no Figure 2 in the uploaded manuscript, so it is not possible to review this text.

L213. Add missing ‰

L219. Perhaps "This is because" instead of "This is while"?

L299-300. Known by whom? Please add references. Also, please comment as to whether this would still hydrolyze organic carbon compounds that are bound within the carbonate lattice.

L322-332. Is this the best proof of concept test? More information is needed on how Mg/Ca was determined as well as the sensitivity of the estimated $\delta^{13}\text{C}_{\text{org-POM}}$ to uncertainties in temperature, calcification DIC, and pCO_2 concentrations.

Reviewer #2 (Remarks to the Author):

This is my second review of Hoogakker et al., in which they present evidence for close tracking of $\text{d}^{13}\text{C-POM}$ by the organic linings and cytoplasm of planktonic foraminifera found in core top and tow samples. I find that the manuscript has been vastly improved and I appreciate the authors' attention to my prior comments. In particular, the authors have added a new data point investigating the signal of foraminifera $\text{d}^{13}\text{C-org}$ from the last glacial maximum. I recognize that this cannot be a true "calibration" exercise given the lack of $\text{d}^{13}\text{C-POM}$ measurements from the LGM, but the authors convincingly show that the value is a reasonable one based on what we expect. The authors also now clearly present the tow samples and core-top samples according to the different types of organic matter they include (i.e. tows include cytoplasm, and it is interesting that the cytoplasm also seems to track POM). I also found the methodological discussion and limitations of sample size to be interesting and clear. Finally, it is also now clear how symbiont bearing versus barren species behave in their basic proxy systematics, confirming expectations. It will be exciting to see this proxy applied and to investigate the nature and diagenesis of organic linings in the future, both for this and for the d^{15}N proxy. Until then, the authors' reference to the Kast et al. paper is well justified. I do not have any suggestions for further improvement of the manuscript.

Point by point response to reviewers' comments:

REVIEWER COMMENTS

Reviewer #1 (Remarks to the Author):

R1 review of Hoogaaker et al., "Planktonic foraminifera organic carbon isotopes as novel archives of upper ocean carbon cycling," submitted to Nature Communications, by Jesse Farmer

I am torn on the revised version of this manuscript. On one hand, the authors' fascinating results remain in place (they have even added one new analysis), and the fundamental motivation and framing of this study remains solid.

On the other hand, there are two major deficiencies in the submitted manuscript. First, since the initial submission, the study of Swart et al. (2021, GCA) was published investigating the $\delta^{13}\text{C}$ of planktonic foraminifera-bound organic matter in the eastern tropical Pacific. I do not believe the existence of the Swart et al. publication by itself fundamentally changes the novelty or significance of this current study. However, it is critical that the authors' compare to the methods and results of Swart et al. throughout for completeness. As it stands, the authors only mention this study toward the end of the manuscript, and here largely to raise a point about methodological differences. This is not sufficient.

We have revised our manuscript to accommodate this comment by the reviewer:

1. In our introductory section we finish with the following sentence: "We also compare our Atlantic results with a recent study by Swart et al.²⁸ from across the tropical equatorial Pacific."
2. The discussion part is thoroughly rewritten.

Second, there are numerous errors in the submitted manuscript that made for a challenging review. There was no Figure 2 in the uploaded manuscript, nor a tracked changes version. However, at least some of the additions mentioned by the authors in the response to reviews are not well integrated with the existing manuscript. For instance, there is no presentation of the new LGM data within the Results section or on any figure (I should note that these does appear later in the discussion). And again, on L269-271, the observation of no difference between cytoplasm and foraminifera-bound organic matter $\delta^{15}\text{N}$ is thrown on the end of this paragraph; it is not clear to the reader how this observation supports the authors' reasoning.

We thank the reviewer for noting these errors in the submitted manuscript and have addressed them in this revised version. We did submit a tracked changes version of the manuscript, and do not know why the reviewer did not have access to this (question to the editor) Furthermore, the journal instructions specifically said only to upload figures that have been revised, hence only Figure 1 was uploaded, as Figure 2 remained unchanged.

We added a line about the new LGM data to our results section.

This was a comment added at the recommendation of the second reviewer. For clarification we have added "compared with POM" at the end of this sentence.

Ultimately, the authors' may be given another opportunity for major revisions to better integrate the new literature on foraminifera-bound organic material $\delta^{13}\text{C}$ with their results and improve the readability and presentation of the manuscript. A cautionary note is that these changes may lead to a longer and more field-specific manuscript that would be better suited to a different journal, but I leave that decision to the editors.

Line-by-line comments.

L47. Suggest change to "the $\delta^{13}\text{C}$ of specific benthic foraminifera".

We have changed this.

L53-56. It would greatly help this paragraph if this proxy-specific point were removed, as this would keep the paragraph focused on big-picture concerns.

We have removed these sentences to focus on the bigger picture.

L58-63. This point could be more strongly made with an actual illustration (a figure). We have rewritten this paragraph.

"While we have some knowledge about modern $\delta^{13}\text{C}_{\text{org-POM}}$, little is known about past $\delta^{13}\text{C}_{\text{org-POM}}$. For example, during ice ages/ glacial stages, lower atmospheric CO_2 concentrations and colder sea surface temperatures are thought to have caused a reduction in air-sea fractionation, which in turn would have caused enrichment in $\delta^{13}\text{C}_{\text{org-POM}}$. Bulk sediment $\delta^{13}\text{C}_{\text{org}}$ measurements from the Atlantic over the last glacial-interglacial cycle however do not show uniform enrichment²³⁻²⁶. This suggests that locally either other factors play a role, or that processes other than pCO_2 /temperature can overprint the signal in bulk sediments²⁷. Here we assess whether planktonic foraminifera-bound $\delta^{13}\text{C}_{\text{org}}$ ($\delta^{13}\text{C}_{\text{org-pforam}}$) provides a better representation of suspended $\delta^{13}\text{C}_{\text{org-POM}}$. We focus on a N-S Atlantic Ocean transect, using living and recent (Holocene) planktonic foraminifera. We also compare our Atlantic results with a recent study by Swart et al.²⁸ from across the tropical equatorial Pacific."

These lines are to emphasize how little we know about past $\delta^{13}\text{C}_{\text{Org-POM}}$. The glacial/interglacial hypothesized difference is just an example. We are not testing the hypothesis of glacial enriched $\delta^{13}\text{C}_{\text{Org-POM}}$ in the manuscript; this would need a basin-wide representation, which we are not presenting here.

The focus of the manuscript is to show that planktonic foraminifera $\delta^{13}\text{C}_{\text{Org}}$ are strikingly similar to $\delta^{13}\text{C}_{\text{Org-POM}}$ and therefore represent an exciting new potential proxy which could be applied in future studies.

L68-86 This section "Distribution of suspended $\delta^{13}\text{C}_{\text{Org-POM}}$ in the Atlantic Ocean" seems out of place and a mash of introduction, results, discussion, and interpretation. I'm not quite sure what to do with this, but perhaps saving it for after the planktonic foraminifera organic carbon and materials & methods sections would help.

We have moved this to earlier in the introduction to follow the text on POM. This section describes what $\delta^{13}\text{C}_{\text{Org-POM}}$ looks like in the modern ocean, followed by the Atlantic, and what factors influence it. It is disconnected from the material and methods, results and discussion section whose main focus is to test whether $\delta^{13}\text{C}_{\text{Org-FOM}}$ represent $\delta^{13}\text{C}_{\text{Org-POM}}$.

L80. Add hyphen "13C-depleted"

Done.

L93: move semi-colon to after "omnivorous": "Planktonic foraminifera are omnivorous; their diet varies between species and habitat, but includes..."

Done.

L120. Swart et al. should also be cited here.

Done.

L143. Add comma after first "species"

Done.

L152. Check this sentence for typo or grammatical error.

We changed "For net-caught planktic foraminifera specimens with cytoplasm, picked under a binocular stereomicroscope, were analyzed."

To "For net-caught samples, planktonic foraminifera specimens with cytoplasm were analyzed."

L154. Add hyphen, e.g. "net-caught"

Done.

L155. Recommend specifying latitude range here e.g., 32 to 62 °N, 41 to 42 °S We

have added this. Note that this section has been moved after the discussion.

L174. What volume of bleach was used? Was any evaluation done to see if there was sufficient

oxidative capacity in the bleach to oxidize all non-bound organic C? For context, Ren et al. used 10 mL of 13% bleach for 6 hours on up to 10 mg of foraminifera.

We used 10 ml of 13% bleach for 4 hours. In the supplementary information file of Ren et al. (2009), they show that for a 5 to 10 mg sample there is 4 $\mu\text{mol/g N}$ (reduced from 17 $\mu\text{mol/g N}$). This would be similar to 0.006% of the sample. If we assume Redfield C to N ratios, then under the same circumstances there would be about 0.03% of C in the sample. Our planktonic foraminifera core samples typically contained just under 0.03% of C, providing confidence that all non-bound organic C was oxidized. We have added a comment reflecting this to the methods section.

L183. HCl

Done.

L188-190. What were the quality control samples? And are the “duplicate samples for $\delta^{13}\text{C}_{\text{Org}}$ ” referring to $\delta^{13}\text{C}_{\text{Org-pforam}}$? Please specify.

We’ve added this line to the methods “The quality control sample is an in-house soil reference, calibrated to NIST standards sucrose and (NIST 8542) and USGS40 (NIST 8573) as normalisation standards.” We have added planktonic foraminifera to be more specific about the duplicate samples.

L203. There is no Figure 2 in the uploaded manuscript, so it is not possible to review this text.

This is because the Figure was not changed from the original manuscript, and the journal requested that only revised Figures are uploaded with the revisions.

L213. Add missing ‰

Done.

L219. Perhaps “This is because” instead of “This is while”?

This is an observation: While both open ocean $\delta^{13}\text{C}_{\text{Org-POM}}$ and sediment $\delta^{13}\text{C}_{\text{Org}}$ are on average enriched compared with continental margins, sediment $\delta^{13}\text{C}_{\text{Org}}$ in the open ocean is further enriched compared with $\delta^{13}\text{C}_{\text{Org-POM}}$. Because would infer a causation.

L299-300. Known by whom? Please add references. Also, please comment as to whether this would still hydrolyze organic carbon compounds that are bound within the carbonate lattice.

We’ve deleted “known”, as this refers to soil organic and wastewater treatments. No research has been carried out on planktonic foraminifera, however various studies on soil organic matter and wastewater have shown that autoclaving causes hydrolysis and changes in chemical structure.

As this reviewer also requested a more in-depth discussion of the Swart et al. (2021), we have rewritten and expanded on this paragraph considerably:

“There are three previous studies looking at $\delta^{13}\text{C}_{\text{Org-pforam}}$, all aiming to use this method as a proxy to reconstruct atmospheric CO_2 (Maslin et al.³⁹ and Swart et al.²⁸ we can compare $\delta^{13}\text{C}_{\text{Org-pforam}}$ with local $\delta^{13}\text{C}_{\text{Org-POM}}$. In both cases, $\delta^{13}\text{C}_{\text{Org-pforam}}$ are depleted compared with local $\delta^{13}\text{C}_{\text{Org-POM}}$. Maslin et al.³⁹, working in the Atlantic, also measured bulk sediment $\delta^{13}\text{C}_{\text{Org}}$ which, while showing similar results to their $\delta^{13}\text{C}_{\text{Org-pforam}}$, appear depleted by $\sim -4.5\%$ compared with our core top compilation in Figure 1. In their study Maslin et al.³⁹ used a dialysis step, and it is likely that this step

removes labile organic material, potentially causing depletion of the remaining $\delta^{13}\text{C}_{\text{Org}}$. The recent work of Swart et al.²⁸, in the Pacific, uses a different approach from Maslin et al.³⁹, with a different instrumental set-up and cleaning protocol. The $\delta^{13}\text{C}_{\text{Org-pforam}}$ results of Swart et al.²⁸ are however also depleted (-7‰) compared with local $\delta^{13}\text{C}_{\text{Org-POM}}$. Swart et al.²⁸ propose that this discrepancy is a result of the chemical composition of foraminifera bound organic material, attributing the depleted $\delta^{13}\text{C}_{\text{Org-pforam}}$ results to a substantial lipid component. This is in contrast with previous studies suggesting that the foraminifera-bound organic material is predominantly composed of polysaccharides and proteins^{29,36,58}. Comparable normalization standards were used in our and Swart et al.²⁸ studies, hence it is unlikely that instrumentation was responsible for the contrasting results. Instead, we take a closer look at the cleaning protocols applied prior to analysis.

In our approach we opted for a methodology that minimizes potential labile organic carbon loss, and potential contamination. Prior to analyses our samples were not exposed to temperatures exceeding 40-50°C and we used acid vapour at room temperature to remove the calcium carbonate tests. We also minimized the introduction of non-laboratory grade/ and non-ultra-pure chemicals following the cleaning steps to limit potential sources of contamination. In their cleaning steps, Swart et al. (2021) introduced two steps that exposed the foraminifera to elevated temperatures: 1- a reductive cleaning test where the sample vials were put in an 80°C bath, and 2- an oxidative cleaning test where samples were autoclaved for 2 hours at 122°C. Autoclaving can cause hydrolysis and has been associated with changes in the chemical structure of soil organic matter and waste water^{59,60}. While the foraminifera test may protect the organic carbon compounds bound within from aqueous cleaning methods, it cannot protect the organic carbon from extreme pressure and temperature changes. Heating the foraminifera to 80/122 °C may cause alteration of labile organic compounds bound to the carbonate lattice, potentially into smaller compounds and/or gaseous losses. Once the calcite test is removed, such gasses would be released. We hypothesize that the depleted $\delta^{13}\text{C}_{\text{Org-pforam}}$ measured by Swart et al.²⁸ are an artefact of the sample treatment, with more labile organic carbon lost due to heating during their cleaning process. We suggest that only an approach that analyses all of the organic components held within foraminifera-bound organic carbon, as applied here, can produce comparable results to $\delta^{13}\text{C}_{\text{Org-POM}}$.

One of the limitations of our approach is the large sample size needed to obtain meaningful analyses. To widen the proxy's application, sample sizes would need to be reduced, which may be facilitated by tailoring the instrumental setup. It would be worthwhile assessing whether the differences between ours and the Swart et al.²⁸ cleaning protocols are predictable, in which case a derivative calibration may be developed to enable the analyses of samples that requires considerably less material."

L322-332. Is this the best proof of concept test? More information is needed on how Mg/Ca was determined as well as the sensitivity of the estimated $\delta^{13}\text{C}_{\text{Org-POM}}$ to uncertainties in temperature, calcification DIC, and pCO₂ concentrations.

We have added an estimate of the total error involved with this calculation:

"With a calcification temperature of 6.6°C (Mg/Ca = 1.2 mmol/mol), *G. bulloides* $\delta^{13}\text{C}_{\text{Calcite}}$ of 0.46‰, and atmospheric CO₂ concentration of 192 ppmv, we estimate that glacial $\delta^{13}\text{C}_{\text{Org-POM}}$ is -18.6±0.7‰, taking into account uncertainties relating to temperature estimates (±1°C, equal to ±0.6‰), $\delta^{13}\text{C}_{\text{DIC}}$ measurement (±0.1‰), and ice core CO₂ (±10 ppmv, equal to 0.3‰)."

Additional changes:

We have now started our discussion with the paragraph of sample timing, as otherwise it is in the middle of two paragraphs that discuss foraminifera bound organic carbon isotopes.

Reviewer #2 (Remarks to the Author):

This is my second review of Hoogakker et al., in which they present evidence for close tracking of d13C-POM by the organic linings and cytoplasm of planktonic foraminifera found in core top and tow samples. I find that the manuscript has been vastly improved and I appreciate the authors' attention to my prior comments. In particular, the authors have added a new data point investigating the signal of foraminifera d13C-org from the last glacial maximum. I recognize that this cannot be a true "calibration" exercise given the lack of d13C-POM measurements from the LGM, but the authors convincingly show that the value is a reasonable one based on what we expect. The authors also now clearly present the tow samples and core-top samples according to the different types of organic matter they include (i.e. tows include cytoplasm, and it is interesting that the cytoplasm also seems to track POM). I also found the methodological discussion and limitations of sample size to be interesting and clear. Finally, it is also now clear how symbiont bearing versus barren species behave in their basic proxy systematics, confirming expectations. It will be exciting to see this proxy applied and to investigate the nature and diagenesis of organic linings in the future, both for this and for the d15N proxy. Until then, the authors' reference to the Kast et al. paper is well justified. I do not have any suggestions for further improvement of the manuscript.

We thank the reviewer for their thorough review and positive comments.